# Synthesizing Libraries of Programs with Auxiliary Functions

**Habibur Rahman**                                            *habibur@ualberta.ca*
*Amii, Department of Computing Science, University of Alberta*

**Thirupathi Reddy Emireddy**                                 *emireddy@ualberta.ca*
*Amii, Department of Computing Science, University of Alberta*

**Kenneth Tjhia**                                             *tjhia@ualberta.ca*
*Amii, Department of Computing Science, University of Alberta*

**Elham Parhizkar**                                           *parhizka@ualberta.ca*
*Amii, Department of Computing Science, University of Alberta*

**Levi H. S. Lelis**                                          *levi.lelis@ualberta.ca*
*Amii, Department of Computing Science, University of Alberta*

**Reviewed on OpenReview:** *https://openreview.net/forum?id=tP1PBrMUlX*

## Abstract

A common approach to program synthesis is to use a learned function to guide the search for a program that satisfies the user's intent. In this paper, we propose a method that offers search guidance, through a domain-dependent auxiliary function, that can be orthogonal to the guidance previous functions provide. Our method, which we call Auxiliary-Based Library Learning (AULILE), searches for a solution in the program space using a base algorithm. If this search does not produce a solution, AULILE enhances the language with a library of programs discovered in the search that optimizes for the auxiliary function. Then, it repeats the search with this library-augmented language. This process is repeated until a solution is found or the system reaches a timeout. We evaluate AULILE in string manipulation tasks. AULILE improved, in some cases by a large margin, the performance of several base algorithms that use different search and learning strategies: BUS, BUSTLE, CROSSBEAM, and BEE SEARCH. Our results suggest that AULILE offers an effective method of injecting domain knowledge into existing systems through a library learning scheme that optimizes for an auxiliary function.

## 1 Introduction

There is increasing interest in the discovery of programmatic hypotheses, i.e., hypotheses encoded as programs written in a domain-specific language (Ellis et al., 2023; Singh & Gulwani, 2015). Depending on the language used, these hypotheses can be interpretable (Medeiros et al., 2022), more amenable to verification (Bastani et al., 2018), and generalize to unseen scenarios (Inala et al., 2020). The key difficulty in synthesizing such hypotheses is the size of the space of the programs. In addition to being large, the program space is often discontinuous, making it difficult to use gradient descent-based optimization algorithms.

A popular solution to speed up the synthesis process of programmatic hypotheses is to learn functions to guide the search of algorithms such as Bottom-Up Search (BUS) (Albarghouthi et al., 2013; Udupa et al., 2013) and Top-Down Search (Wang et al., 2017). BUSTLE (Odena et al., 2021), PROBE (Barke et al., 2020),

BEE SEARCH (Ameen & Lelis, 2023), and CROSSBEAM (Shi et al., 2022) enhance BUS with a guiding function. In addition to learning a guiding function, DREAMCODER (Ellis et al., 2023) also learns a library of programs while solving a set of training tasks. DREAMCODER uses a compression scheme to learn a set of programs that can be reused as part of other hypotheses. Despite all these advances, depending on the size of the programmatic solution, existing methods still struggle to synthesize effective programs.

In this paper, we consider the scenario in which one is able to encode domain knowledge through an auxiliary function that can be helpful in guiding the search for programmatic hypotheses. We use the auxiliary function to learn task-specific libraries of programs. That is, instead of learning libraries of programs for a problem domain, like DREAMCODER does, we learn a library of programs by searching for a solution to a specific task. We use a base synthesizer to search for a solution to the task. If the search is not successful, we augment the language with the program encountered in the search that optimizes for the auxiliary function, i.e., the program becomes a function in the language. The search for a solution is then repeated with the augmented language. Since the program added to the language can be used through a simple function call, the second search might still not solve the problem, but it may find yet another program that uses the program inserted into the language in the previous iteration that better optimizes the auxiliary function. We hypothesize that the auxiliary function will guide the search through a sequence of augmentation steps that transforms the synthesis of a complex solution into an easy task, where the last search of the base synthesizer combines the programs added to the language. We refer to this process as Auxiliary-Based Library Learning (AULILE).

We also hypothesize that, since the auxiliary function is likely different from previously learned guiding functions, AULILE can provide search guidance that is orthogonal to that provided by existing functions. That is, using an auxiliary function can increase the number of problems existing systems can solve. To test this hypothesis, we used AULILE to augment BUS, BUSTLE, BEE SEARCH, and CROSSBEAM in string manipulation tasks (Alur et al., 2013; Odena et al., 2021). AULILE improved the performance of all synthesizers in terms of the number of tasks solved, in some cases by a large margin, thus supporting our hypothesis.

The empirical results we present in this paper suggest that AULILE offers an effective way to inject domain knowledge into the synthesis of programmatic hypotheses. The results also suggest future directions for systems learning how to guide the search in programmatic spaces, as a simple auxiliary function can already improve the performance of existing systems; would learned auxiliary functions provide even better guidance?

Our implementation is available online.[1]

## 2 Problem Formulation

In program synthesis, one searches the space of programs defined by a domain-specific language (DSL), which is provided in the form of a context-free grammar $\mathcal{G}$. The DSL includes a set of non-terminals ($V$), terminals ($\Sigma$), and relations ($R$) that define the production rules and the initial symbol of the grammar ($I$). Figure 1 (left) illustrates a DSL where $V = \{I\}$, $\Sigma = \{\texttt{<},\texttt{>}, i_1, \ldots, i_k, \texttt{concat}, \texttt{replace}\}$, and $R$ defines the production rules (e.g. $I \rightarrow \texttt{<}$). A production rule with at least one non-terminal symbol on the right-hand side is referred to as non-terminal, while a production rule with no non-terminal symbols on the right-hand side is referred to as terminal. The set of programs $\mathcal{G}$ accepts defines the program space. For instance, the program $\texttt{replace(concat(<, >), <, >)}$ is accepted by $\mathcal{G}$: we start with $I$ and replace it with $\texttt{replace}\,(I, I, I)$; then we replace the leftmost $I$ with $\texttt{concat}(I, I)$, the middle non-terminal $I$ with $\texttt{<}$, and the rightmost non-terminal with $\texttt{>}$, etc.

Programs are represented as abstract syntax trees (ASTs). Figure 1 (right) illustrates the AST of the program $\texttt{replace(concat(<, >), <, >)}$. Every node in the AST represents a production rule. For example, the root of the AST in Figure 1 represents $I \rightarrow \texttt{replace}(I, I, I)$. Nodes representing a non-terminal rule have a number of children corresponding to the number of non-terminal symbols in the rule. Nodes corresponding to terminal rules are leaves in the AST. Note that each subtree in the AST represents a program. A subtree rooted at a child of node $p$ is referred to as a subprogram of $p$. For instance, $\texttt{concat(<, >)}$, $\texttt{<}$, and $\texttt{>}$ are subprograms of $\texttt{replace(concat(<, >), <, >)}$.

---

[1]https://github.com/lelis-research/aulile

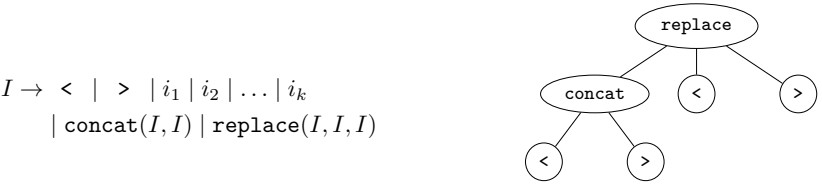

$$I \rightarrow \texttt{<} \mid \texttt{>} \mid i_1 \mid i_2 \mid \ldots \mid i_k$$
$$\mid \texttt{concat}(I, I) \mid \texttt{replace}(I, I, I)$$

Figure 1: DSL and AST for `replace(concat(<,>),<,>)`.

A program synthesis task consists of a DSL $\mathcal{G}$, a set of inputs $\mathcal{I} = \{I_1, \ldots, I_n\}$, and a set of outputs $\mathcal{O} = \{O_1, \ldots, O_n\}$. Here, each input $I_i$ and output $O_i$ can each represent a set of values. The goal is to find a program $p \in \mathcal{G}$ that correctly maps each input set to its corresponding output set, $p(I_i) = O_i$, for $i = 1, \ldots, n$. For example, `concat`$(i_1, i_2)$ solves $\mathcal{I} = \{[\texttt{<},\texttt{>}], [\texttt{>><},\texttt{>}], [\texttt{>},\texttt{>}]\}$ and $\mathcal{O} = \{[\texttt{<>}], [\texttt{>><>}], [\texttt{>>}]\}$, where $i_1$, and $i_2$ are the input values associated with each input $I_i$.

## 3 Base Synthesizers Used in Our Study

In this section, we describe the synthesizers used in our augmentation study: bottom-up search (BUS) (Albarghouthi et al., 2013; Udupa et al., 2013), BUSTLE (Odena et al., 2021), BEE SEARCH (Ameen & Lelis, 2023), and CROSSBEAM (Shi et al., 2022). All synthesizers used in our experiments are based on the BUS algorithm. We chose to use these methods because they represent the current state of the art and all represent improvements over BUS through a learned guiding function. By using our augmentation method with all of them, we can measure how much our enhancement can improve existing methods from the literature, i.e., how much the guidance of our approach is orthogonal to the guidance of existing approaches.

In the following sections, we describe the synthesizers used in our study in a level of detail that allows the reader to understand how the augmentation approach is implemented in each of them. We refer the reader to the original papers of each method for a more detailed description of the approaches.

### 3.1 Bottom-Up Search (BUS)

BUS incrementally builds programs of increasing size. The process begins by generating all programs defined by the terminal symbols of the DSL, which have size 1 in terms of the number of nodes in the program's AST. Then, using the programs of size 1, BUS generates the programs of size 2 by applying the production rules. This process continues using the programs of sizes 1 and 2 to generate programs of size 3, and so on. The search stops when a program is generated that can map the inputs to the outputs of the task.

Consider a scenario in which we need to synthesize a program that produces the output `><>` with the DSL of Figure 1 (left). The set of inputs for this task is empty and a solution is `concat(concat(>, <), >)`. To find the solution, BUS begins by generating and evaluating all programs of size 1, which are `>` and `<` in this case. Since neither of these programs correctly generates the output, BUS proceeds to generate the set of programs of size 2, which is empty in this example, because the programs of size 1 cannot be combined into programs of size 2 in this DSL. Next, BUS generates all programs of size 3, including `concat(<, <)`, ..., `concat(>, >)`. This process continues until the correct program is generated. This example illustrates how BUS performs a systematic search to find a solution by increasing the size of the programs it generates.

One of the key features of the synthesizers we use in our study is their ability to perform observational-equivalence checks. If $p_1$ and $p_2$ produce the same outputs for all inputs in $\mathcal{I}$, we can keep $p_1$ or $p_2$ and discard the other. Observational-equivalence can dramatically reduce the size of the effective search space.

### 3.2 Bustle

BUSTLE performs a cost-guided bottom-up search. Instead of enumerating programs in terms of AST size, as BUS does, BUSTLE enumerates them in terms of cost. Here, the cost of a program is determined by a

cost function, which in BUSTLE's case is a trained neural network. Cheaper programs are deemed to be more likely to be part of a solution to the problem and are evaluated first in the search.

BUSTLE's cost function employs a neural network to compute the likelihood that a program is a subprogram of a solution program. The network receives the input-output pairs $(\mathcal{I}, \mathcal{O})$ and the outputs $p(I)$, $I \in \mathcal{I}$, of a program $p$ and returns the probability that $p$ is part of a solution. This neural cost function is determined through two functions: $w$ and $w'$. Let $p = r(p_1, \cdots, p_k)$ be a program determined by the production rule $r$ whose $k$ non-terminal symbols are replaced by programs $p_1, \cdots, p_k$, the $w$-value of $p$ is computed as follows:

$$w(p) = 1 + \sum_{i=1}^{k} w'(p_i).$$

The value 1 represents the cost of the rule $r$ and $w'(p_i)$ is the cost of the program $p_i$ as given by the following:

$$w'(p) = w(p) + 5 - \delta(p), \tag{1}$$

where $\delta(p) \in \{0, \cdots, 5\}$ is an integer that is based on the probability of $p$ being part of a solution. The value of $\delta(p)$ is computed by binning the probability value the model returns. BUSTLE uses the values in $\{0.0, 0.1, 0.2, 0.3, 0.4, 0.6, 1.0\}$ as follows: if the probability returned by the model is within the first two values $[0.0, 0.1)$, $\delta(p) = 0$, if it is within the second and third values $[0.1, 0.2)$, $\delta(p) = 1$, etc. $\delta$ penalizes $p$ according to the probability the neural network assigns to $p$; lower probabilities result in higher costs.

### 3.3 Bee Search

BEE SEARCH also performs a cost-based search in the space of programs. However, in contrast to BUSTLE, it does not require the cost function to produce integer values. Ameen & Lelis (2023) introduced a variant of the BUSTLE cost function, denoted $w_{\mathrm{U}}$, which does not use the binning scheme to discretize the cost values:

$$w_{\mathrm{U}}(p) = 1 + \sum_{i=1}^{k} w'_{\mathrm{U}}(p_i) \tag{2}$$

where,

$$w'_{\mathrm{U}}(p_i) = w_{\mathrm{U}}(p_i) - \log_2 \mathbb{P}(p_i).$$

Similarly to BUSTLE's $w$, Equation 2 assumes that the cost of each production rule is 1 and that the $w_{\mathrm{U}}$-cost of a program $p$ is given by 1 plus the sum of the $w'_{\mathrm{U}}$ costs of the subprograms of $p$. The difference between BUSTLE's $w$-function and $w_{\mathrm{U}}$ is that the latter penalizes programs $p$ with $\mathbb{P}(p)$, the model's estimated likelihood of $p$ being part of a solution, and not with a scheme that bounds the penalty to at most 5.

### 3.4 Crossbeam

Similarly to BUS, BUSTLE, and BEE SEARCH, CROSSBEAM keeps all generated programs in memory so it can perform observational-equivalence checks. However, instead of generating all combinations of existing programs with a production rule $r$ as BUS does, CROSSBEAM samples subprograms $p_i$ from the existing set of programs to determine which programs $r(p_1, \cdots, p_k)$ are generated and evaluated next in search.

CROSSBEAM trains a model that receives all the programs generated in the search so far, the input-output pairs, and a production rule $r$; the model produces a probability distribution over all the programs encountered in the search to define what the programs $p_i$ should be that comprise the next program $r(p_1, \cdots, p_k)$ to be evaluated. The sampling procedure for programs $p_i$ can be performed with Beam Search or UniqueRandomizer (Shi et al., 2020). If the sampled program $p$ is observational-equivalent to a previously seen program, then $p$ is discarded. Crossbeam iteratively goes through all production rules $r$ and samples programs $r(p_1, \cdots, p_k)$, until a solution is found or the search times out.

---

**Algorithm 1** AULILE

---

**Require:** Input-output pairs $\mathcal{I}, \mathcal{O}$, DSL $\mathcal{G}$, auxiliary function $a$, base synthesizer $Y$, and a budget $B$
**Ensure:** Solution program $p$ or $\perp$
1: **while** not timeout **do**
2:     $p$, solved $\leftarrow Y(\mathcal{G}, \mathcal{I}, \mathcal{O}, B, a)$
3:     **if** solved **then**
4:         **return** $p$
5:     $\mathcal{G} = \mathcal{G} \cup p$
6: return $\perp$

---

## 4 Auxiliary-Based Library Learning

Our language augmentation method based on an auxiliary function, AULILE, is presented in Algorithm 1. We denote the augmented version of a base synthesizer $Y$ as A-$Y$. For example, the augmented version of BUS is denoted A-BUS. AULILE receives a problem specification through input-output pairs $\mathcal{I}$ and $\mathcal{O}$, a DSL $\mathcal{G}$, an auxiliary function $a$, a base synthesizer $Y$, and a computational budget $B$. AULILE returns either a solution program $p$ or a failure signal $\perp$ as output. AULILE repeatedly invokes the base synthesizer $Y$ to explore the space of programs (line 2). Each search with $Y$ is bounded by a computational budget $B$ (e.g., number of programs evaluated in the search). If the base synthesizer finds a solution, then $p$ represents this solution (line 4). Otherwise, $p$ represents the program encountered in the search that optimizes the auxiliary function $a$, in which case $p$ is added to the language so that it can be used in the next iteration of the search. The base synthesizer is configured to return a program $p$ that was not added to the language in previous iterations of AULILE. Different base synthesizers handle the addition of $p$ to the language differently. We discuss how the synthesizers used in our experiments handle language augmentation in Sections 4.1 and 4.2. AULILE runs while there is available time for synthesis.

### 4.1 A-BUS, A-Bustle, and A-Bee

The program $p$ added to the language in A-BUS, A-BUSTLE, and A-BEE is treated as a non-terminal symbol of the type that $p$ returns. For example, if $p$ returns an integer value, then it can be used as an operation that returns an integer. This means that all added programs have a cost of 1 (size 1 in the context of BUS), so they are evaluated early in the search and can be more easily combined with other programs.

### 4.2 A-Crossbeam

In the context of CROSSBEAM, the program $p$ used to augment the language is added to the initial set of programs explored by the search. When sampling the next program $p'$ to be evaluated in the search, CROSSBEAM will be able to use $p$ as one of the subprograms of $p'$ starting in the first iteration of the search.

The model CROSSBEAM uses to create the probability distribution on existing programs is trained on-policy. This means that CROSSBEAM uses the distribution of programs seen during the search to solve training problems to train the model. Since the programs $p$ added to the language can be of arbitrary complexity, it is unlikely that the CROSSBEAM model has trained on search contexts similar to those A-CROSSBEAM induces (i.e. the set of existing programs might contain complex programs even in the first iteration of search). We empirically evaluate if CROSSBEAM's model is able to generalize to the search contexts of A-CROSSBEAM.

Since CROSSBEAM uses sampling, it can benefit from random restarts (Hoos & Stützle, 2004). That is, instead of running the system with a computational budget of $X$ program evaluations, if we use $N$ random restarts, we would sequentially perform $N$ independent runs of the system with a budget of $\frac{X}{N}$ program evaluations each. Since A-CROSSBEAM implicitly performs random restarts, where in each restart it uses an augmented version of the language, it would not be clear from the results if performance improvements were due to random restarts or augmentation. Therefore, we also use a version of CROSSBEAM that uses random restarts without augmentation as a baseline in our experiments, denoted by CROSSBEAM(N), where $N > 1$.

```
P1(arg):                                              P2(arg):
    arg.replace("-", "")                                  P1(arg)
        .replace(".", "")                                     .replace("<", "")
        .replace(
            concat(">", " "), "")
```

| Input | P1's Output |
|-------|-------------|
| 801-456-8765 | 8014568765 |
| <978> 654-0299 | <9786540299 |
| 978.654.0299 | 9786540299 |

| Input | P2's Output |
|-------|-------------|
| 801-456-8765 | 8014568765 |
| <978> 654-0299 | 9786540299 |
| 978.654.0299 | 9786540299 |

Figure 2: Example of the augmentation process in a string manipulation task.

### 4.3 Example of A-BUS on the String Manipulation Domain

Consider the example shown in Figure 2. The table on the right-hand side represents both the input-output pairs of the task and the output of a solution program, P2. Although the solution program is long and search algorithms such as BUS would not be able to find it due to computational limitations, A-BUS with a simple auxiliary function finds a solution. In the first iteration of A-BUS, it adds to the language program P1, which does not solve the problem, but produces output strings that are closer to the output strings of the problem specification. Among all generated programs, P1 is the one that better optimizes the auxiliary function (defined in Section 5). In the next iteration, the solution P2 uses P1 as a subprogram.

### 4.4 Weaknesses of Aulile

The main weakness of AULILE is its requirement of an auxiliary function. In this paper, we evaluate search algorithms in string manipulation problems, for which there exists an obvious auxiliary function—the Levenshtein distance (see Section 5). However, there might be no obvious choices for other domains. Future work will investigate how one can learn auxiliary functions to augment the language. AULILE also inherits some of the weaknesses of its base synthesizer. For example, BUS algorithms are memory-intensive, and so will be AULILE if using a BUS algorithm as its base synthesizer. Stochastic local search algorithms (Husien & Schewe, 2016b; Medeiros et al., 2022) can explore repeated programs because they do not keep a list of evaluated programs. AULILE will suffer from the same problem if using such algorithms as base synthesizer.

## 5 Empirical Evaluation

We evaluate AULILE on string manipulation (Alur et al., 2013; Odena et al., 2021) tasks, in which one is given a set of input-output examples and needs to find a program that maps each input to its corresponding output. We use two datasets from Odena et al. (2021): one with 89 instances of the SyGuS competition and another with 38 handcrafted instances. We implemented the DSL for string manipulation, as well as BUS and BUSTLE. We use the implementations of CROSSBEAM and BEE SEARCH provided by their authors.

We perform three sets of experiments. In the first set, we compare the base synthesizers with their augmented counterparts (Figures 3 and 4). We compare the algorithms in terms of the number of problems solved by the number of programs evaluated. This is because all algorithms use the same model as a cost function and thus have a similar per-evaluation computational cost. In the second set, we compare the best performing systems of each experiment in the first set (Figure 5). Since the algorithms use different models, we use running time instead of number of evaluations. This is to account for the complexity of the models employed. For example, CROSSBEAM uses a more complex and thus slower model than BUSTLE. We used 14 million programs evaluated as the budget $B$ of AULILE for A-BUS, A-BUSTLE, and A-BEE.

In the third set, we perform ablation studies. In AULILE, if the base synthesizer does not solve the problem, it adds to the language the program $p$ with the best $a$-value that is different from the programs added in previous iterations of AULILE. We evaluate a version of A-BUS that adds to the language, in each iteration,

the best $k \geq 1$ programs that are different from the programs added in previous iterations, for $k$ in $\{1, 2, 3\}$ (Figure 6). We also present results of A-BUS with auxiliary functions of varied strengths (Figure 7).

The budget $B$ is determined differently for A-CROSSBEAM due to the restarts we implemented with CROSSBEAM. In the first set of experiments, given a computational budget $X$, which is denoted by the rightmost point on the x-axis of our plots, we present curves for runs of CROSSBEAM and A-CROSSBEAM with different values of $N$. For a given $N$, the algorithm restarts after $\frac{X}{N}$ evaluations. For legibility, CROSSBEAM is abbreviated as `CB` and A-CROSSBEAM as `A-CB` in the plots; the values in brackets denote $N$. In the second set of experiments, we used 5 million evaluations as $B$ for A-CROSSBEAM. We use a smaller budget with A-CROSSBEAM because, as we show below, the algorithm can benefit not only from the augmentation of the language, but also from search restarts due to its sampling process. This is in contrast to A-BUS, A-BUSTLE, and A-BEE, which are deterministic and thus tend to benefit from the largest possible $B$.

We plot the mean and standard deviation over five independent runs for BUSTLE, A-BUSTLE, BEE SEARCH, A-BEE, CROSSBEAM, and A-CROSSBEAM because they depend on the random initialization of the weights of their neural models and also sampling in the case of CROSSBEAM and A-CROSSBEAM. BUS and A-BUS are deterministic, therefore we present the results of a single run. We used 1 CPU at 2.4 GHz and 64 GB of RAM in all experiments; for BUSTLE, A-BUSTLE, BEE SEARCH, A-BEE, CROSSBEAM, and A-CROSSBEAM we also used 1 GPU, since they use a neural model.

**Auxiliary Function**   We use the Levenshtein distance (Levenshtein, 1966) as the auxiliary function. The Levenshtein distance between two words is the minimum number of single-character edits (insertions, deletions, or substitutions) required to change one word into the other. The higher the distance, the greater the dissimilarity between the strings. The auxiliary function measures the sum of the distances between the output of a program $p(I_i)$ and its corresponding output $O_i$, for all $I_i \in \mathcal{I}$. If two or more programs have the same Levenshtein distance, we arbitrarily select one of them to be added to the language in Algorithm 1.

## 5.1   Empirical Results: First Set

Figure 3 shows the results for the SyGuS benchmark, while Figure 4 shows the results for the 38 benchmark set. Each figure shows a plot for a base synthesizer: BUS, BUSTLE, BEE SEARCH, and CROSSBEAM. The y-axis shows the total number of solved problems, and the x-axis shows the number of programs evaluated. For example, A-BUS solves 87 problems after evaluating $5 \times 10^8$ programs. The line `CB` without brackets represents CROSSBEAM as presented in its original paper (Shi et al., 2022).

The auxiliary-based language enhancement increased the number of problems solved by all base synthesizers evaluated on the SyGuS benchmark, often by a large margin. For example, it increased the number of problems BUS solved from 74 to 87. The results of CROSSBEAM show that not only the augmentation approach is effective, but also simply performing restarts already increases the number of problems the system can solve. Namely, CROSSBEAM with no restarts solved an average of 66.2 problems, while CROSSBEAM with 4 restarts solved an average of 69.0 problems. The best A-CROSSBEAM, `A-CB(8)`, solved an average of 75.6 problems. Improvements to the 38 benchmark problems are more modest than those observed on the SyGuS benchmark, but the augmented approach is never worse and sometimes superior (e.g., A-CROSSBEAM).

## 5.2   Empirical Results: Second Set

In Figure 5 we compare the best performing algorithms from Figures 3 and 4 in terms of running time, rather than the number of programs evaluated. We allowed the same computational budget in terms of running time (x-axis) for both SyGuS and the 38 benchmarks. Note that the results shown in Figure 5 are not directly comparable to those shown in Figures 3 and 4. This is because the computational budget used in the first set of experiments can be different from the budget used in this set. This is particularly noticeable for A-CROSSBEAM, which uses a much smaller computational budget in Figure 4 than in Figure 5.

The results show that A-BUSTLE and A-BEE are the best performing systems on the SyGuS benchmark, while A-CROSSBEAM is the best performing on the 38 benchmark. Interestingly, A-BUSTLE and A-BEE solved almost all instances of the SyGuS benchmark, and A-CROSSBEAM solved all instances of the 38

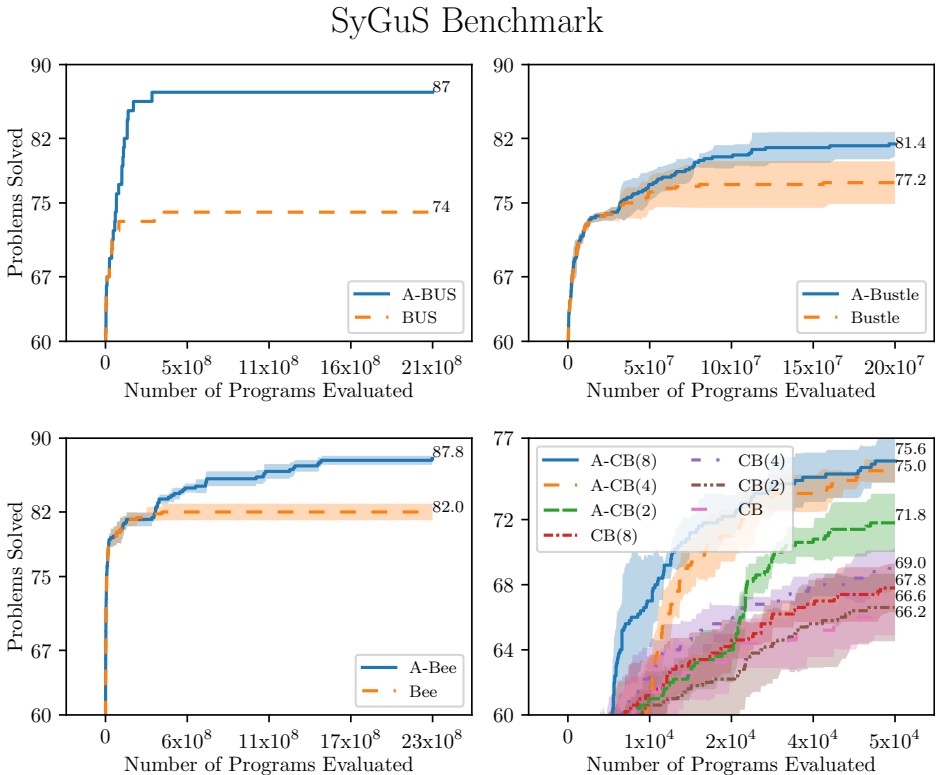

Figure 3: Number of problems solved per number of evaluations on the SyGuS benchmark for four base synthesizers: BUS, Bustle, Bee Search, and Crossbeam. This benchmark set has 89 problems.

benchmark. These results suggest that the community may need more difficult string manipulation benchmarks. Although A-BUS is not the best performing system on the SyGuS benchmark, it is notable that it performs so well on this set of problems. While A-Bustle, A-Bee, and A-Crossbeam employ a neural model to guide the search, A-BUS simply relies on the uninformed search of BUS and the augmentation of the language with the auxiliary Levenstein function.

### 5.3  Empirical Results: Third Set

Figure 6 shows the results of A-BUS when it adds the best $k$ programs to the language in each Aulile iteration, for $k$ in $\{1, 2, 3\}$. The results on the SyGuS benchmark show that there is no variance in the number of problems solved for the small values of $k$ used in this experiment. The results on the 38 benchmark show a small advantage for $k = 3$, as A-BUS with that number solves 33 problems, while it solves 32 with $k = 1$ and $k = 2$. This experiment shows that Aulile is robust to small values of $k$ in these benchmarks.

Figure 7 shows the number of problems A-BUS can solve per number of programs evaluated when using auxiliary functions of different strengths. We define different auxiliary functions that are based on the Levenshtein distance. This is achieved with a parameter $0.0 < l \leq 1.0$. The value of $l$ determines the percentage of the longest of the two strings used to compute the distance. For example, for $l = 0.50$, we compute the Levenshtein distance considering only half of the characters of the longest of the two strings. If $l = 1.0$, then this metric is exactly the Levenshtein distance. Values of $l < 1.0$ result in weaker versions of the distance because it considers less information. As we weaken the auxiliary function by decreasing the value of $l$, A-BUS solves fewer problems, thus showing the importance of the auxiliary function used.

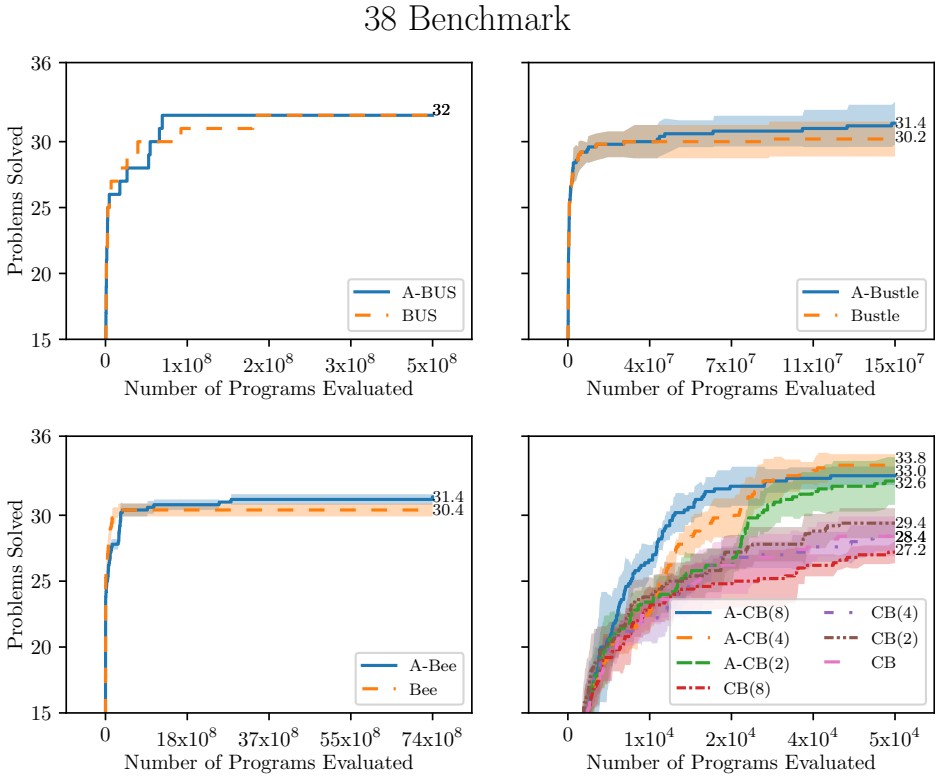

Figure 4: Number of problems solved per number of evaluations on the 38 benchmark set for four base synthesizers: BUS, BUSTLE, BEE SEARCH, and CROSSBEAM. This benchmark set has 38 problems.

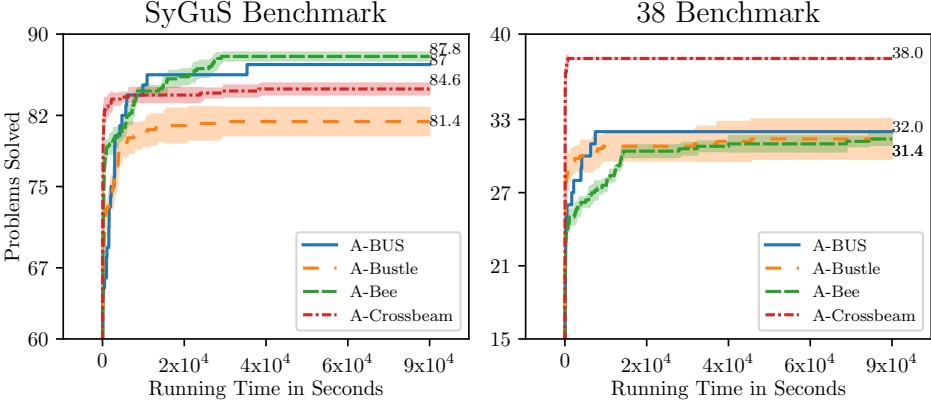

Figure 5: Number of problems solved per running time on the SyGuS and the 38 benchmark sets for the best performing system for each base synthesizer, as shown in Figures 3 and 4.

### 5.4 Discussion

The results of our experiments suggest that our auxiliary-based language augmentation approach offers search guidance that is orthogonal to that provided by existing methods. This is because it improved the results of BUSTLE, BEE SEARCH, and CROSSBEAM. While BUSTLE and BEE SEARCH use similar cost functions to guide the search, CROSSBEAM uses an entirely different approach. The models of BUSTLE and BEE SEARCH are conditioned on the problem to be solved, while CROSSBEAM's is conditioned not only on the problem,

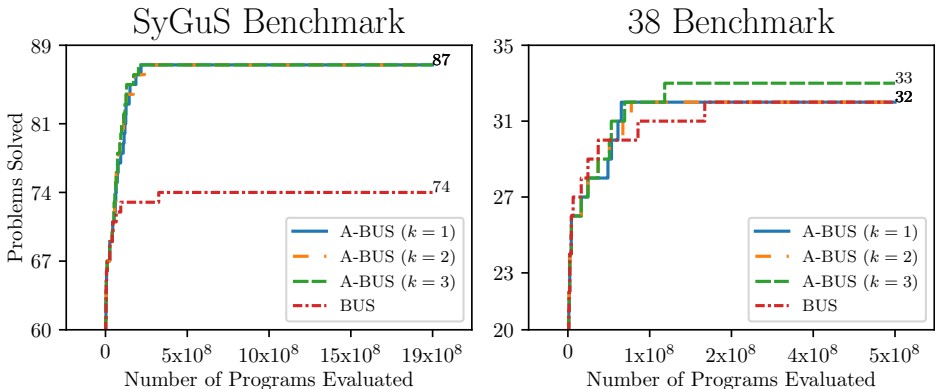

Figure 6: Number of problems solved per number of programs evaluated on the SyGuS and on the 38 benchmark sets for A-BUS with different number $k$ of programs added to the language in each AULILE loop.

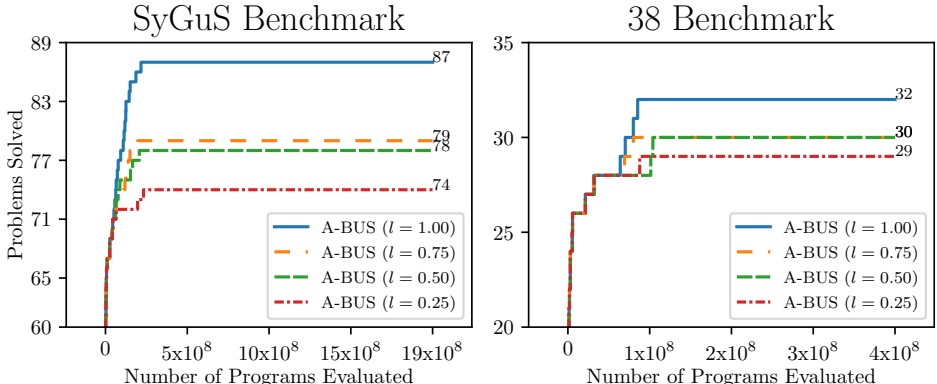

Figure 7: Number of problems solved per number of programs evaluated on the SyGuS and the 38 benchmark sets for A-BUS with an auxiliary function of different strengths, where $l = 1.00$ is the original Levenshtein distance; values $l < 1.00$ represent weakened versions of the function.

but also on the search state (i.e., the set of programs evaluated in search). Despite these differences, our augmented approach improved the results of all three systems, in addition to the uninformed BUS.

## 6 More Related Works

The challenge of synthesizing computer programs that meet a given specification has been widely discussed in the field of Computing Science (Manna & Waldinger, 1971; Summers, 1977), drawing significant attention from researchers in Artificial Intelligence (Balog et al., 2016; Devlin et al., 2017a; Kalyan et al., 2018; Ellis et al., 2023) and Programming Languages (Lee et al., 2018; Barke et al., 2020; Ji et al., 2020).

A variety of search methods have been investigated to solve synthesis tasks. One such method is to use constraint satisfaction algorithms, which convert the synthesis task into a constraint satisfaction problem that can be solved with an SMT solver (Solar-Lezama, 2009). Additionally, synthesis tasks can be addressed using stochastic search algorithms, such as Simulated Annealing (Husien & Schewe, 2016a; Medeiros et al., 2022), stochastic hill climbing (Aleixo & Lelis, 2023), and genetic algorithms (Koza, 1992). In this paper, we evaluate the augmented approach with algorithms based on BUS, which are enumerative since they systematically evaluate a set of programs in the space defined by the DSL. Top-down search algorithms are also enumerative, but search in a tree where the root is given by the initial symbol of the grammar defining the language; the children of a node in the tree are given by the different ways of how a non-terminal symbol

can be replaced by other symbols of the grammar. The key disadvantage of top-down algorithms is that all internal nodes of the search tree are partial programs that cannot be executed, thus only allowing weaker forms of equivalence checks (Lee et al., 2018; Wang et al., 2017). The ability of removing observational-equivalent programs justifies our choice for BUS-like algorithms. However, our approach can be used with stochastic, top-down Chen et al. (2019); Zohar & Wolf (2018); Bunel et al. (2018); Devlin et al. (2017b); Wang et al. (2017), and other bottom-up base synthesizers (Barke et al., 2020; Fijalkow et al., 2022).

DREAMCODER also learns a library of programs, which involves compressing the programs used to solve a set of training problems (Ellis et al., 2023). Our method differs from DREAMCODER in important ways. First, DREAMCODER learns a library of programs for a problem domain given a set of training tasks. AULILE learns a library of programs for a specific task, without training tasks and no reuse of the library across tasks. Second, DREAMCODER learns a model to guide the search given a library of programs, while our approach either does not use a model to guide the search (e.g., A-BUS) or simply uses existing models without retraining for the learned library (e.g., A-BUSTLE, A-BEE, and A-CROSSBEAM). DREAMCODER was extended in two different ways. STITCH is a search-based system aimed at improving the performance of DREAMCODER's compression system in terms of memory and running time (Bowers et al., 2023). BABBLE improves the set of learned programs by considering the functionality of the programs despite their potential syntactical differences (Cao et al., 2023). AULILE differs from STITCH and BABBLE because it does not use compression to learn a library of programs, but uses an auxiliary function to learn a task-specific library.

## 7 Conclusions

In this paper, we introduced Auxiliary-Based Library Learning (AULILE), a system that leverages a domain-specific auxiliary function to learn task-specific libraries of programs. AULILE uses a base synthesizer to search for a programmatic solution to a task; in the case of failure, AULILE adds to the language the program encountered in the search that best optimizes the auxiliary function. The search is then repeated with the augmented language. The augmentation and search steps are repeated until one of the following conditions is met: the task is solved, or the system reaches a time out. We evaluated AULILE on string manipulation tasks. Our results suggest that a simple auxiliary function offers guidance that is orthogonal to that provided by existing functions. This is because AULILE was able to improve the performance of all base synthesizers evaluated, in some cases by a large margin. In general, our empirical results suggest that AULILE can offer an effective way to inject domain knowledge into the synthesis process.

A promising direction for future research is to investigate learning schema for the auxiliary function. We envision two directions for learning auxiliary functions. The first is to investigate the use of existing cost functions as auxiliary functions (Odena et al., 2021; Barke et al., 2020). Can the cost function be used effectively to select the program that is added to the language in AULILE? The second direction is to try to learn auxiliary functions that provide orthogonal information to what existing cost functions provide.

### Acknowledgments

This research was supported by Canada's NSERC and the CIFAR AI Chairs program, and was enabled in part by support provided by the Digital Research Alliance of Canada. The authors thank the reviewers for their helpful suggestions.

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

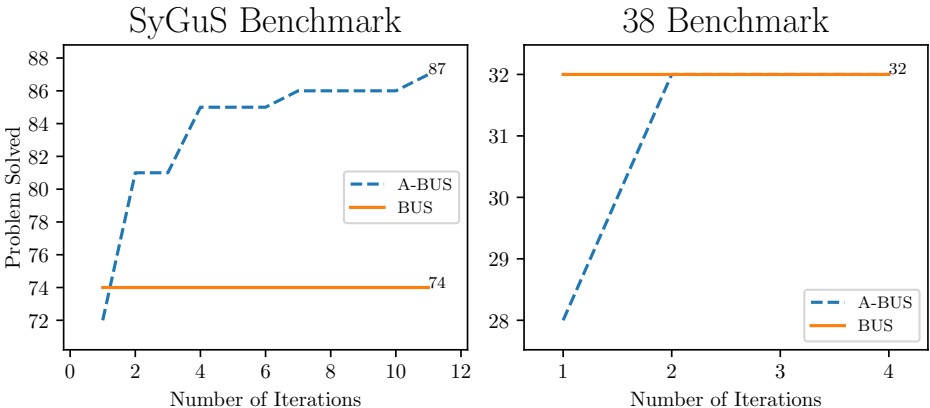

Figure 9: Number of problems solved per number of iterations of A-BUS. BUS is shown with a flat horizontal line because it does not perform language augmentation.

# A    Appendix

## A.1    DSL for String Processing Domain

| Expression, E | $\rightarrow$ | $S \mid I \mid B$ |
|---|---|---|
| String expression, S | $\rightarrow$ | $\mathrm{Concat}(S1, S2) \mid \mathrm{Left}(S, I) \mid \mathrm{Right}(S, I) \mid \mathrm{Substr}(S, I1, I2)$ |
| | | $\mid \mathrm{Replace}(S1, I1, I2, S2) \mid \mathrm{Trim}(S) \mid \mathrm{Repeat}(S, I) \mid \mathrm{Substitute}(S1, S2, S3)$ |
| | | $\mid \mathrm{Substitute}(S1, S2, S3, I) \mid \mathrm{ToText}(I) \mid \mathrm{LowerCase}(S) \mid \mathrm{UpperCase}(S)$ |
| | | $\mid \mathrm{ProperCase}(S) \mid T \mid X \mid \mathrm{If}(B, S1, S2)$ |
| Integer expression, I | $\rightarrow$ | $I1 + I2 \mid I1 - I2 \mid \mathrm{Find}(S1, S2) \mid \mathrm{Find}(S1, S2, I) \mid \mathrm{Len}(S) \mid J$ |
| Boolean expression, B | $\rightarrow$ | $\mathrm{Equals}(S1, S2) \mid \mathrm{GreaterThan}(I1, I2) \mid \mathrm{GreaterThanOrEqualTo}(I1, I2)$ |
| String constants, T | $\rightarrow$ | "" $\mid$ " " $\mid$ "," $\mid$ "." $\mid$ "!" $\mid$ "?" $\mid$ "(" $\mid$ ")" $\mid$ "[" $\mid$ "]" |
| | | $\mid$ "<" $\mid$ ">" $\mid$ "" $\mid$ "" $\mid$ "-" $\mid$ "+" $\mid$ "_" $\mid$ "/" $\mid$ "\$" $\mid$ "#" |
| | | $\mid$ ":" $\mid$ ";" $\mid$ "@" $\mid$ "%" $\mid$ "0" $\mid$ string constants extracted from I/O examples |
| Integer constants, J | $\rightarrow$ | $0 \mid 1 \mid 2 \mid 3 \mid 99$ |
| Input, X | $\rightarrow$ | $x_1 \mid \ldots \mid x_k$ |

Figure 8: DSL considered for the string manipulation domain.

## A.2    Problems Solved per Iteration of Language Augmentation

Figure 9 shows the cumulative number of problems A-BUS can solve in each iteration of the search for both the SyGuS and the 38 benchmarks. Although most of the problems are solved in the first iteration of the search (72 for SyGuS and 28 for the 38 benchmark), many problems are solved with more iterations. In particular, one of the problems in the SyGuS benchmark is solved after 10 iterations of the AULILE's language augmentation loop. We note that the number of problems solved in the first iteration of A-BUS does not match the number of problems BUS solves due to the limitation of A-BUS's budget $B$—each iteration of A-BUS is limited to evaluating 14 million programs, while the only iteration of BUS evaluates 2.1 billion programs. The gap we observe between A-BUS and BUS (87 versus 74 problems solved) highlights the advantage of augmenting the language with programs that optimize the score of the auxiliary function.

## A.3    Aulile's Average Program Size

Figure 10 shows the average program size of the solution program that A-BUS finds in different language augmentation iterations. The average program size, in terms of the number of nodes in the AST of the program, is calculated for all solution programs encountered in a given iteration of A-BUS. The plot also

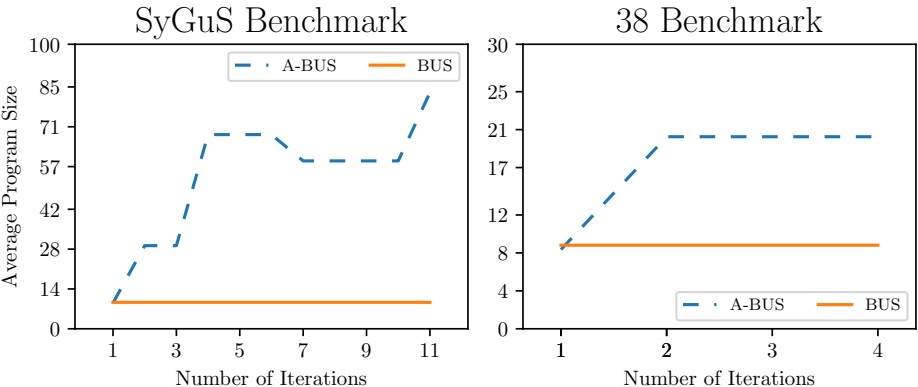

Figure 10: Average program size for the problems solved across different iterations of A-BUS.

presents the average solution program size BUS finds. The size of the solution programs grows rapidly in the early iterations (1–4 for the SyGuS benchmark and 1 and 2 for the 38 benchmark), but the average size becomes somewhat constant in both domains in later iterations. The sizes of the ASTs of the solution programs A-BUS finds are much larger than those of the solution programs BUS finds. Although BUS is optimal with respect to the AST size, it is unable to solve many of the problems in the two benchmarks evaluated. AULILE offers the option of trading the simplicity of the solution for more problems solved.

