# OpenReview forum: "Synthesizing Libraries of Programs with Auxiliary Functions"
_TMLR — Accepted by TMLR_

### Review · Reviewer_e3NF · 2023-11-21

**Summary Of Contributions:**

This paper proposes to augment methods for program synthesis from input-output specifications based on bottom-up search, by additionally using an auxiliary function to help select among incomplete programs to reuse in later iterations of the search. The proposed method, Aulile, invokes an inner search method in a loop. In each iteration, the inner method returns either a solution, or the program found with the best score according to the auxiliary function. This program is added to the set of base programs for the next iteration of the outer loop.

The authors evaluate their approach added to 4 different bottom-up search methods, using two sets string manipulation tasks used in prior work. They show that it can improve the success rate compared to the baseline method.

**Audience:**

Yes

**Broader Impact Concerns:**

I have no concerns about broader impact.

**Claims And Evidence:**

Yes

**Requested Changes:**

I would like to see the authors try at least one more auxiliary function and analyze how that impacts the performance of the method, in order to answer the following questions:
- How does the choice of auxiliary function affect the performance of the method?
- Can we see whether the use of a better auxiliary function is correlated with increased performance on the method?

Some examples could include the difference in length between the program's output and the true output, and the length of the prefix which matches between the program's output and the true output.

**Strengths And Weaknesses:**

Strengths:
- The proposed method is quite simple and easy to understand.
- The proposed method shows empirical improvement on benchmarks compared to prior work.

Weaknesses:
- The generality of the method is unclear as the paper only considers one program domain (string manipulation). It may be not so straightforward to pick a good auxiliary function for other domains.
- By adding large programs as primitives to the bottom-up search, equal to all the other regular primitives, the method may end up generating more complex programs than otherwise.
- The paper proposes to use the auxiliary function provided in the outer loop of the process, but does not investigate other ways to use the same information. Bustle, Bee Search, and Crossbeam all contain a component to learn whether some program fragment is likely to be useful as part of the ultimate solution. Using an auxiliary function which explicitly computes Levenshtein distance could be a way to shortcut this learning, and might work better inside the inner loop as well.

---

> ### Author Response · Authors · 2024-02-02
> **Responses to Review**
>
> Thank you for your thoughtful review and suggestions!
>
> Following your suggestion, we have added the results of an experiment where we evaluate ABUS with auxiliary functions of different strengths. This was achieved by computing the Levenshtein distance using prefixes of the longest string, as suggested. Please see Figure 7 of the revised paper. ABUS is able to solve fewer and fewer problems as we make the prefix shorter and shorter. This result highlights the role of the auxiliary function. Thank you for suggesting it.
>
> We have also added a plot showing how the complexity of the solution programs change across different iterations of ABUS. Indeed the programs ABUS finds are more complex than those BUS finds. This is a similar tradeoff we observe in heuristic search algorithms such as WA* and Greedy Best-First Search (GBFS). They find solutions of lower quality than A* with an admissible heuristic function. However, in many scenarios, the solutions WA* and GBFS find are all we have, since A* simply fails to find any solution. We believe we are observing something similar here, where we lose optimality in terms of AST size (as BUS guarantees, for example), but we are able to solve many more problems.
>
> Thank you for the other suggestions such as using an auxiliary function in the inner loop. We believe that both using the auxiliary function in the inner loop and the learned functions in the outer loop are interesting research questions to be investigated in future work.

---

### Review · Reviewer_ATes · 2023-11-27

**Summary Of Contributions:**

This paper presents Auxiliary-Based Library Learning (Aulile) for program synthesis from input-output examples, which can be combined with various base program synthesis algorithms. Basically, after each synthesis iteration, if the task is not solved yet, Aulile picks one synthesized program that is the closest to the target outputs, then adds that program into the library, so that in the next iteration, the synthesizer can directly call that program as a subprogram. They evaluate Aulile in the string manipulation domain, including SyGuS benchmark and a collection of 38 tasks from prior work. They demonstrate that Aulile can improve base program synthesizers by a notable margin on SyGuS.

**Audience:**

Yes

**Broader Impact Concerns:**

No concerns.

**Claims And Evidence:**

Yes

**Requested Changes:**

1. Add information on how many iterations are conducted for the algorithm to finish in general, and what is the number of solved problems after each iteration.

2. Add an analysis on the complexity of programs solved in each iteration.

3. Explain the differences between the setting of Figure 5 and Figures 3-4, and why the results are different in the 2 sets of plots.

4. In the design of Algorithm 1, add discussion on whether adding multiple subprograms improve or decrease the sample efficiency.

5. Explain whether it is required to do a retraining for those algorithms that involve neural networks after each iteration, and the additional training cost.

**Strengths And Weaknesses:**

Strengths:
The proposed approach is simple, and show consistent improvement when combined with multiple base program synthesis algorithms on SyGuS.

Weaknesses:
In general, this work lacks ablation studies and error analysis to understand where the performance improvement comes from.

1. In Algorithm 1, in general how many iterations are conducted for the algorithm to finish, and what is the number of solved problems after each iteration?

2. Is there any analysis on the complexity of programs solved in each iteration? For example, the improvement of Aulile might be because using the subprograms enables the algorithm to find more complex programs after each iteration, but it is unclear from the current results.

3. What are the differences between the setting of Figure 5 and Figures 3-4? I wonder why the results are different in the 2 sets of plots.

4. In the design of Algorithm 1, have the authors tried adding multiple subprograms in each iteration? Does it improve or decrease the sample efficiency?

5. After adding new subprograms in the search space, is it required to do a retraining for those algorithms that involve neural networks? What is the additional training cost?

---

> ### Author Response · Authors · 2024-02-02
> **Responses to Review**
>
> Thank you for your thoughtful review and suggestions!
>
> 1. We have added a plot to the Appendix of the revised paper (see Figure 9) where we show the number of problems ABUS can solve per iterations.
> 2. We added a plot showing the size of the AST of the solution program per iterations to the Appendix (see Figure 10).
> 3. We added the following sentence to Section 5.2:
>
> *Note that the results shown in Figure 5 are not directly comparable to those shown in Figures 3 and 4. This is because the computational budget used in the first set of experiments can be different from the budget used in this set. This is particularly noticeable for A-Crossbeam, which uses a much smaller computational budget in Figure 4 than in Figure 5.*
>
> 4. We added experiments showing that ABUS is equally efficient when adding 1, 2, or 3 programs to the language in each iteration (see Figure 6 of the revised version).
> 5. We don't perform any re-training. We use the models as they are. We touch on this topic in Section 4.2, when we discuss this possible issue for A-Crossbeam:
>
> *Since the programs $p$ added to the language can be of arbitrary complexity, it is unlikely that the Crossbeam model has trained on search contexts similar to those A-Crossbeam induces (i.e. the set of existing programs might contain complex programs even in the first iteration of search). We empirically evaluate if Crossbeam's model is able to generalize to the search contexts of A-Crossbeam.*

---

### Review · Reviewer_UZNi · 2024-01-08

**Summary Of Contributions:**

The paper tackles the problem of synthesizing solution programs for a given set of input-output examples. It presents a new method for program synthesis using search-based guidance through a pre-defined domain-dependent auxiliary function, called AULILE. The method operates by building on top of existing search-based program synthesis methods such as Bottom-Up Search (BUS), BUSTLE, BEE search, and CROSS BEAM (referred to as the base synthesizer) by optimizing for the pre-defined auxiliary function in addition to the search-based (optimization) methods of the base synthesizer. To empirically verify the benefit of the AULILE, the authors use a string manipulation task and augment AULILE on base synthesizers using BUS, BUSTLE, BEE, and CROSSBEAM. The results show that methods augmented with AULILE outperform the base synthesizers on both string manipulation benchmarks SyGuS (significant gains) and a hand-crafted benchmark consisting of 38 instances.  The methods also perform reasonably well in terms of their runtime. The auxiliary function used for these tasks was the Levenstein distance between the output of the found program and the expected output for the specific string-based test case.

**Audience:**

Yes

**Broader Impact Concerns:**

In its current state, the paper does not present any ethical concerns.

**Claims And Evidence:**

Yes

**Requested Changes:**

The paper is largely clearly written. However, there are still some additional clarifications required:
- As discussed in the Weaknesses, further clarification on the differences between Dream-Coder and AULILE might be beneficial. Are there specific instances where AULILE performs better, for instance in terms of faster runtimes? It still seems like, Dream-Coder is the more general version of AULILE. Can the string manipulation task be slightly modified to use Dream-Coder as a baseline for AULILE?
- Adding a stochastic top-down program synthesizer to the set of baseline methods would enhance the baselines used in the experiments and showcase the diversity of AULILE further.
The addition of an explicit future work section would be insightful.
- Expanding the scope of AULILE: Currently, the benefits of AULILE are illustrated in the string manipulation domain. Illustration of the method on a slightly more intricate/complex program synthesis domain/DSL or an additional DSL (such as Bit Vector DSL) might strengthen the paper.
The addition of an explicit section on the limitations of the presented approach would make the findings of the paper more comprehensive.

**Strengths And Weaknesses:**

Strengths:
- **Clarity of writing and comprehensive related work**: The paper is written clearly and the baseline methods are discussed in adequate detail given that they are based on related work. The discussion of related work is also comprehensive.
- **Presentation of results**: The results presented on the two benchmarks for the string manipulation task were clear and addressed the hypothesis adequately.

Weaknesses:
- **Domain dependency of the auxiliary function**: I see the benefit of having a domain-dependent function to further guide the search of the solution program. However, finding the optimal auxiliary function for different domains can be challenging. For the string manipulation task, used in the experiments, the Levenstein distance function was appropriate, but for more complex domains finding such a function may not be trivial.
- **In-depth analysis of auxiliary function**: Given the critical role of the auxiliary function in enhancing the search for solution codes, it would be valuable to explore the impact of different auxiliary functions when combined with various base synthesizers.
- **Additional experiments with Top-down search algorithms**: The paper briefly mentions AULILE's use with stochastic top-down base synthesizers. Expanding on this aspect by including the results from this synthesizer alongside others would improve the diversity of the reported outcomes.
- **Comparison with DREAM-CODER**: The paper discusses the differences between the presented approach with Dream Coder; specifically, that Dream Coder requires training tasks to learn a library of programs for a problem domain while AULILE doesn't require training tasks and learns programs in a task-specific manner. However, this still doesn't explain why one should prefer AULILE over Dream Coder, and in which instances would the former be the better choice. Further clarification on this point might be insightful.
- **Lack of discussion on future work**:  For a more organized structure, it might be beneficial to reserve a dedicated section to discuss potential directions for future research.

---

> ### Author Response · Authors · 2024-02-02
> **Responses to Review**
>
> Thank you for your thoughtful review and suggestions. We have addressed your comments to the best of our ability. Please see the detailed comments below.
>
> **Re: In-Depth Analysis of Auxiliary Function**
>
> Thank you for suggesting the addition of experiments evaluating the impact of the auxiliary function. We have added experiments where we evaluate ABUS with weakened versions of the Levenshtein distance. The results are shown in Figure 7 of the revised version. In short, ABUS solves fewer and fewer problems as we weaken the auxiliary function.
>
> **Re: Adding Results with Top-Down Approaches**
>
> We appreciate the suggestion and we agree that it would be interesting to evaluate AULILE with top-down methods. However, previous work [1, 2] showed that bottom-up approaches are the state-of-the-art in the benchmarks we considered in our study. That is why we focused on BUS-based methods. Note that nothing in AULILE is specific to bottom-up approaches, so we would expect it to also enhance top-down methods (enumerative or stochastic local search).
>
> **Re: DreamCoder as a Baseline**
>
> We discussed quite extensively whether we should invest time and resources on having DreamCoder as a baseline in our experiments. The settings in which the two algorithms are used are so different that we decided to not include such a comparison. DreamCoder is more general in the sense that it doesn't require an auxiliary function as input, since everything is learned from data. This makes DreamCoder more applicable to problems in which an effective auxiliary function isn't readily available. However, DreamCoder requires training problems to learn its library of programs. Another factor in favor of AULILE is its simplicity. ABUS is just multiple iterations of BUS with a simple auxiliary function.
>
> **Re: Discuss Future Work**
>
> We have added a paragraph discussing promising directions. Please see the last paragraph of the paper.
>
> **Re: Add a Weaknesses Section**
>
> We added a section on the weaknesses of AULILE (see the Section 4.4 of the revised paper). In this section we discuss the need of an auxiliary function and how AULILE will inherit some of the weaknesses of its base synthesizer.
>
> **References**
>
> [1] Augustus Odena, Kensen Shi, David Bieber, Rishabh Singh, Charles Sutton, and Hanjun Dai. BUSTLE: Bottom-up program synthesis through learning-guided exploration. In International Conference on Learning Representations, 2021. URL https://openreview.net/forum?id=yHeg4PbFHh.
>
> [2] Shraddha Barke, Hila Peleg, and Nadia Polikarpova. Just-in-time learning for bottom-up enumerative synthesis. Proceedings of the ACM on Programming Languages, 4(OOPSLA):1–29, 2020.

---

### Decision · Action_Editor_sPyb · 2024-03-04

**Recommendation:** Accept as is

**Comment:**

The advances made in this paper and fairly narrow in technique (using a domain-specific auxiliary function to identify promising program subroutines) and domain, but the claims seem appropriately scoped. Most of the requests by reviewers for additional experiments were addressed. In total, the matching of evidence and claims is appropriate for TMLR.

**Audience:**

Yes, there is a contingent of the TMLR audience interested in program synthesis, and this paper adds to the area in ways that would appeal to people working directly on related problems.

**Claims And Evidence:**

Yes. The main unaddressed issue raised in reviews is a comparison to dreamcoder. While I agree with the reviewer that it would be a nice addition, I don't think that it is necessary to support the key claims in the paper (that the proposed approach is simpler but requires domain knowledge in place of an analogous dreamcoder training step).